# Multi-Wavelength Photoacoustic Temperature Feedback Based Photothermal Therapy Method and System

**DOI:** 10.3390/pharmaceutics15020555

**Published:** 2023-02-07

**Authors:** Yiming Ma, Yang Liu, Zhigang Lei, Zezheng Qin, Yi Shen, Mingjian Sun

**Affiliations:** 1Department of Control Science and Engineering, Harbin Institute of Technology, Harbin 150000, China; 2WEGO Holding Co., Ltd., Weihai 264209, China; 3Department of Control Science and Engineering, Harbin Institute of Technology, Weihai 264209, China

**Keywords:** photothermal therapy, multi-wavelength, temperature feedback control

## Abstract

Photothermal therapy (PTT) is a new type of tumor treatment technology that is noninvasive, repeatable, and does not involve radiation. Owing to the lack of real-time and accurate noninvasive temperature measurement technology in current PTT surgical procedures, empirical and open-loop treatment laser power control mode inevitably leads to overtreatment. Thermal radiation causes irreversible damage to normal tissue around cancer tissue and seriously affects the therapeutic effect of PTT and other therapies conducted at the same time. Therefore, real-time measurement and control of the temperature and thermal damage of the therapeutic target are critical to the success of PTT. To improve the accuracy and safety of PTT, we propose a multi-wavelength photoacoustic (PA) temperature feedback based PTT method and system. PA thermometry information at different wavelengths is mutually corrected, and the therapeutic light dose is regulated in real time to accurately control the treatment temperature. The experimental results on the swine blood sample confirm that the proposed method can realize real-time temperature measurement and control of the target area with an accuracy of 0.56 °C and 0.68 °C, demonstrating its good prospects for application.

## 1. Introduction

Photothermal therapy (PTT) is a new treatment technology that uses external laser sources to convert light energy into heat to kill cancer tissue in the lesion [1,2,3,4,5,6,7,8,9,10]. PTT has the advantages of enabling local treatment, limiting damage to normal tissue, high selectivity, and high treatment efficiency, and it has become a promising technology in the field of tumor clinical treatment. Because PTT uses the difference in temperature tolerance between normal tissue and tumor tissue to kill diseased cells, accurate temperature measurement and control are very relevant for its safety and effectiveness in terms of thermal damage. However, in current PTT operations, owing to the lack of non-invasive and high-precision thermometry methods, the treatment laser power choice is mostly based on empirical and open-loop methods. This limitation can lead to overtreatment caused by the improper therapy laser energy. Mild photothermal therapy (MPTT) technology is considered a potential solution to the abovementioned problems [11]. In this type of therapy, the treatment temperature of the target area is generally stabilized below 45 °C. Therefore, the key to the success of the MPTT method lies in accurate measurement and control of therapy temperature.

Temperature measurement methods for biological tissue can be divided into noninvasive and interventional methods. The interventional temperature measurement method has the risk of tumor metastasis, which limits its application in tumor MPTT. Therefore, high-precision noninvasive temperature measurement methods have become popular research topics. The main noninvasive temperature measurement methods include infrared, magnetic resonance, and ultrasonic temperature measurements. Infrared temperature measurement is relatively mature at present, but it can only detect the surface temperature of the tissue and cannot yield temperature information of a deep target, which is mainly the result of light scattering by the tissue [12,13]. Although nuclear magnetic resonance temperature measurement can achieve accurate noncontact temperature measurement, it has high equipment cost and large amount of data calculation, which is not conducive to dynamic regulation of the therapy temperature [14,15]. Ultrasonic thermometry technology uses ultrasonic imaging parameters to estimate temperature, which meets the requirements of real-time measurement and detection depth, but the temperature measurement precision is not high enough [16,17,18]. Owing to the lack of noninvasive thermometry methods, the treatment light dose and temperature of PTT are difficult to control. Therefore, empirical and open-loop power regulation methods are used in current tumor PPT operations, and these need to be improved in terms of accuracy and convenience.

The photoacoustic (PA) effect is based on the generation of ultrasonic waves through laser radiation under thermal and stress confinements. With the rapid development of PA imaging, photoacoustic temperature measurement (PATM) technology has gradually attracted attention [19,20,21,22,23,24,25]. The PA effect depends on thermalelastic transformation, and the mechanism of photo-induced ultrasound makes it easier to combine with MPTT technology. However, in its practical application, it is very dependent on the sensitivity of the measurement transducer, which makes the robustness of the temperature measurement system poor and limits its application in clinical PTT surgery. Therefore, it is necessary to combine high-precision temperature measurement with robust temperature control to achieve high-precision PTT.

First, we propose the multi-wavelength PA-excitation-based temperature control method (MPA-TCM) to realize therapeutic laser dose feedback control of the region of interest (ROI). This method extracts the temperature information from the PA signals excited by the multi-wavelength lasers, and the mutual correction and fusion of various information further improves the accuracy and robustness of temperature control. Second, a multi-wavelength photoacoutic temperature feedback based photothermal therapy (MPA-PTT) system with integrated hand-held therapeutic probe is proposed. To verify the functions of the proposed system, we performed a series of in vitro experiments based on a swine blood sample. A discussion of potential solutions to improve the accuracy and security of MPTT is also gprovided.

## 2. Materials and Methods

### 2.1. PA Temperature Measurement

The PA effect refers to the phenomenon through which tissue absorbs modulated light energy and generates sound waves through thermalelastic transformation based on satisfying the temporal stress confinement [19,20]. The initial sound pressure p(z) excited by a short pulse laser can be expressed as:(1)$$p(z)=\Gamma (T)\mu_{a}F(z,\mu_{a},\mu_{s},g,\lambda )+N$$
where F is the incident laser fluence, $\mu_{a}$ is the optical absorption coefficient, $\mu_{s}$ is the scattering coefficient, g is the anisotropic factor, λ is the laser wavelength, and *N* is the random measurement noise. In particular, Γ(T) is the Grüneisen coefficient, which has been experimentally shown to vary linearly with temperature T in the body temperature range [24]:(2)$$\Gamma (T)=\frac{\beta c^{2}}{C_{p}}=\mathrm{KT}+B$$
where β is the thermal expansion coefficient, *c* is the sound velocity, $C_{p}$ is the heat capacity under constant pressure, and the fitting of K and B will also be affected by the use of different wavelengths of lasers for PA excitation [25]. In particular, the temperature dependence of the optical absorption coefficient is assumed to be negligible compared to the temperature dependence of the Grüneisen parameter [23]. Therefore, the PA signal excited at the fixed wavelength λm can be derived as:(3)$$p_{\lambda m}(z)=(\mathrm{KT}+B)\mu_{a}F(z,\mu_{a},\mu_{s},g,\lambda m)+N$$

If $T_{0}$ is set as the original tissue base temperature, then $p(z,T_{0})$ is the PA signal at this temperature:(4)$$p_{\lambda m}(z,T_{0})=(KT_{0}+B)\mu_{a}F(z,\mu_{a},\mu_{s},g,\lambda m)+N$$

Through proportional operation, one can obtain the following:(5)$$T_{\lambda m}=\frac{B+KT_{0}}{Bp_{\lambda m}(z,T_{0})}p_{\lambda m}(z)-\frac{B}{K}=Rp_{\lambda m}(z)+L+E_{\lambda m}$$
where R and L are parameters that can be obtained by PATM at a fixed wavelength λm, and $E_{\lambda m}$ is the abstract form of the deviation in the PA temperature measurement, which is caused by noise *N* and calculation. Thus, the temperature of the target area can be obtained by recording and analyzing the PA information. However, because of the existence of measurement noise and environmental interference, the accuracy of PATM signals obtained at a single optical wavelength is limited. To tackle this issue, we propose the multi-wavelength PA temperature measurement method.

The problem of multi-wavelength PA temperature measurement can be regarded as a multiple linear regression problem. A multiple linear regression algorithm was used to fuse the temperature information generated at different laser wavelengths, thereby improving the accuracy of the temperature measurement and control. Assuming that the PA signal is obtained at *m* excitation wavelengths, the temperature measurement results can be obtained as [23,25], and the fitted weight factor vector *W* is set as:(6)$$W={\left[\begin{array}{ccccc} w_{1} & w_{2} & \ldots & w_{m} & w_{0} \end{array}\right]}^{⊤}$$
(7)$$T(z)=f(p(z,\lambda 1),\ldots ,p(z,\lambda m))=w_{1}p(z,\lambda 1)+\ldots +w_{m}p(z,\lambda m)+w_{0}+\sum_{i=1}^{m}\frac{E_{\lambda i}}{m}$$
where f represents the multiple linear regression mapping. If the PA signal is excited by *m* fixed wavelength lasers at *n* fixed sampling temperatures from the base temperature $T_{0}$ to $T_{0}+(n-1)$. Then, the calibration temperature matrix $T_{a}$ and data matrix X can be expressed as:(8)$$T_{a}={\left(\begin{array}{cccc} T_{0} & T_{0}+1 & \ldots & T_{0}+n-1 \end{array}\right)}^{⊤}$$
(9)$$X=\left[\begin{array}{cccccc} x_{11} & x_{21} & x_{31} & \ldots & x_{m1} & 1\\ x_{12} & x_{22} & x_{32} & \ldots & x_{m2} & 1\\ \ldots & \ldots & \ldots & \ldots & \ldots & 1\\ x_{1n} & x_{2n} & x_{3n} & \ldots & x_{\mathrm{mn}} & 1 \end{array}\right]$$

To find the optimal fit weighting factor vector, we define the square loss function of the model as:(10)$$S\left(W\right)=T_{a}{}^{⊤}T_{a}-2W^{⊤}X^{⊤}T_{a}+W^{⊤}X^{⊤}\mathrm{XW}$$

Differentiating the loss function gives the extreme value used to determine the optimal weighting factor vector $\tilde{W}$:(11)$$\frac{\partial S\left(W\right)}{\partial W}=-2X^{⊤}T_{a}+2X^{⊤}\mathrm{XW}=0$$
(12)$$\tilde{W}={\left(X^{⊤}X\right)}^{-1}X^{⊤}T_{a}={\left[\begin{array}{ccccc} \tilde{w_{1}} & \tilde{w_{2}} & \ldots & \tilde{w_{m}} & \tilde{w_{0}} \end{array}\right]}^{⊤}$$

The pixel value in the PA image is linearly proportional to the pressure increase caused by the PA effect due to the linear nature of the imaging system. The concept of photoacoustic thermometry factor (PATF) $S_{\lambda m}$ is proposed in this paper, which refers to the average intensity value of *a* points in the ROI of the PA image excited by *b* pulsed lasers with wavelength λm, and $V_{\lambda ,i,j}$ is the value of pixels at the corresponding point.
(13)$$S_{\lambda m}=\sum_{i=1}^{b}\sum_{j=1}^{a}\frac{V_{\lambda m,i,j}}{a\cdot b}\;$$
where *b* is an adjustable parameter in the imaging system, and the larger the parameter *b* is, the more acquired PA frames are averaged during the imaging; the imaging quality is better, but the imaging speed will be slowed accordingly. By fitting the linear relationship between the PATF and the temperature using the multiple linear regression algorithm, the temperature information in the pixel values of the PA images with different wavelengths can be extracted using
(14)$$T(z)=f(S_{\lambda 1},S_{\lambda 2},\ldots ,S_{\lambda m})$$

### 2.2. Multi-Wavelength PA-Excitation-Based Temperature Control

Based on PA temperature measurement, MPA-TCM was proposed to regulate the temperature of the target area (Figure 1). First, the reconstruction from PA signals of different wavelengths to PA images was completed. To ensure the rapidity and real-time of signal processing, PA image reconstruction was completed using the filtered back-projection reconstruction algorithm built into the programmable ultrasound imaging system (Prodigy, S-sharp, Taiwan). Then, the temperature information was extracted using multivariate linear fitting, and finally, the closed-loop temperature control was completed. This method combines PA temperature measurement information of different wavelengths and various complementary information to further improve measurement and control accuracy. The temperature image distribution could be obtained by mapping the points in the ROI according to the fitting temperature results. In particular, to ensure the rapidity of temperature control and the effect of temperature imaging, the temperature imaging process was started only after the system reached the stable control stage.

To control the ROI temperature in PTT, the PATF is also used to calculate the required feedback temperature information for MPA-TCM. As shown in Figure 2, the MPA-TCM adopts a cascade PID control strategy, which is divided into an external temperature control loop and an internal voltage control loop. In particular, the sampling frequency of the entire system is determined by the repetition frequency of the pulsed laser.

where $T_{d}(t)$ is the expected value of the target temperature during the PTT process, $e_{1}(t)$ and $e_{2}(t)$ are the error signals of the inner and outer loops, respectively, $U_{r}(t)$ and $U_{d}(t)$ are the control variables of the temperature and voltage loop, respectively, $T_{m}(t)$ is the measured temperature based on the multi-wavelength PA temperature measurement, and $U_{m}(t)$ is the working voltage of the CW laser measured by the measuring circuit. In a control cycle of the MPA-TCM, the above variables are calculated as shown in Equations (15)–(18). The inner and outer loops use a proportional–derivative and proportional–integral control strategy, respectively, where $\alpha_{1}$ and $\alpha_{2}$ are the proportional coefficients of the outer and inner loops, $\beta_{i}$ is the integral coefficient, and $\beta_{d}$ is the differential coefficient. The introduction of a proportional integral derivative (PID) control strategy can ensure the dynamic and static characteristics of the system. In addition, each wavelength of the laser is sequentially excited in a control cycle; finally, the feedback temperature value is calculated through the weight factor matrix.



(15)
$$e_{1}(t)=T_{d}(t)-T_{m}(t)$$


(16)
$$e_{2}(t)=U_{r}(t)-U_{m}(t)$$


(17)
$$U_{d}(t)=\alpha_{1}\left[e_{1}(t)+\beta_{d}\cdot \frac{d(e_{1}(t))}{d(t)}\right]$$


(18)
$$U_{r}(t)=\alpha_{2}\left[e_{2}(t)+\frac{1}{\beta_{i}}\cdot \int_{0}^{t}e_{2}(t)\mathrm{dt}\right]$$



## 3. Material and Methods

### 3.1. Construction of the MPA-PTT System

The proposed MPA-PTT system (Figure 3) was assembled based on a programmable ultrasound imaging system (Prodigy, S-sharp, Hong Kong, China), which was mainly composed of a multi-wavelength optical module, a signal processing and timing control module, a temperature imaging module and a temperature control module. The PA signals were converted into PA and temperature images, and the temperature signals were fed back to the temperature control module in real time to regulate the power of the CW laser. The synchronization of the general system was controlled by the signal processing and timing control module.

For PA temperature imaging and PTT, we utilized an optical parametric oscillation (OPO) pulse laser (SpitLight-600, Innolas, Munich, Germany) with multi-wavelength tunable functionality and a CW laser (LWIRL808-7W-F, Laserwave, Beijing, China) with a wavelength of 808 nm (Figure 4). The temperature control module calculated and adjusted the ROI temperature for a better therapy effect. The pulsed and CW lasers were coupled into a customized double optical outlet fiber. The three-dimensional printed integrated holder (Figure 5 and Figure 6) housed the ultrasonic probe (L7.5, S-sharp, Hong Kong, China) with the fiber and ensured that the lasers could converge at the optimal distance. The optical outlet length of the fiber bundle was the same as the size of the linear transducer array of the ultrasonic probe to ensure the precision of the PTT and the temperature measurement (Figure 5). The numerical aperture of the optical fiber was 0.22, while its diameter was 192 um. In particular, the center frequency of the probe was 7.5 MHz.

The working sequence of the MPA-PTT system is shown in Figure 7. Under the excitation of pulsed laser pumping, the temperature value of the target area was calculated once every 20 cycles of photoacoustic excitation and fed back to the temperature control module to calculate the ideal working power of the CW laser in the next control cycle, thereby ensuring that the temperature of the target area can be adjusted in a real-time closed loop. During the operation of the MPA-PTT system, the number and corresponding wavelength of laser pulses were adjusted via programming.

### 3.2. Sample Preparation

To verify the functions of the MPA-PTT system, we made an ex vivo swine blood sample (Figure 8). The swine blood sample was wrapped in an agar-based phantom, which was created using a common method for tissue mimicking material [26]. The phantom was created by dissolving 8 g of agar and 6 g of gelatin in 100 mL of water and adding 4 g of cornstarch to simulate acoustic scattering. The depth of treatment is the key parameter of PTT and the distance from the swine blood sample to the upper surface of the model was established as 8 mm (Figure 7).

### 3.3. Multi-Wavelength PA Temperature Calibration and Measurement

To verify the multi-wavelength PA temperature measurement function of the proposed system, a multi-wavelength PA temperature calibration and measurement experiment was performed. PA information at wavelengths of 860 and 960 nm was fused to complete the experiment. In addition, the pulse width of the laser was 7 ns and the repetition frequency was 10 Hz. A thermocouple (TT-K30SLE, OMEGA, New York, USA) was inserted into the clot and the power of the CW laser was manually adjusted to increase the temperature, while the heating interval was 30 to 45 °C and the minimum sampling unit was 0.5 °C. The PA information of ROI was repeatedly excited 100 times at each sampling temperature, while the PATF was calculated and used to perform multi-wavelength PA temperature measurement. In particular, the PA temperature information based on a single wavelength (860 and 960 nm) was also calibrated and compared with the PA temperature measurement results based on multiple wavelengths.

### 3.4. Closed-Loop Photothermal Therapy

To verify the closed-loop PTT temperature control function of the proposed system, a simulated closed-loop PTT experiment was performed. The initial temperature of the swine blood sample was 33 °C, while the desired PTT temperature was 42 °C. Based on the multiwavelength (860 and 960 nm) PA temperature information, the ideal power of the CW laser (808 nm) was regulated in real time. To ensure the safety of the experiment, the working powers of the pulsed and CW lasers met the safety requirements of the American National Standard Institute. Real-time multimodal photoacoustic, ultrasonic, and temperature images provided the system operator with more detailed target area information. Control-group experiments based on single-wavelength and multi-wavelength laser information were also conducted to demonstrate the advantages of MPA-TCM.

### 3.5. Anti-Interference Ability Test

To verify the robustness of the proposed system, an anti-interference ability test was performed to reduce the potential external interference affecting the stability and safety of PTT. After the system reached the stable state of temperature control, another CW laser (LWIRL808-7W-F, Laserwave, China) was used to irradiate the sample with a constant heating energy density (8 mJ/cm^2^) to simulate nonrepetitive interference. The self-adjustment ability of the MPA-PTT system under the influence of the external interference heat source was analyzed, and the experimental results based on single-wavelength and multiwavelength lasers were also compared.

## 4. Results

### 4.1. Multi-Wavelength PA Temperature Calibration and Measurement

The relationship between PA information and local temperature of the swine blood sample was extracted to test the feasibility of multi-wavelength PA thermometry. The PA temperature calibration results for the 860- and 960-nm lasers are shown in Figure 9. Under the excitation of two different lasers, the PATF of the swine blood sample exhibited a trend of increasing with increasing temperature, which can be clearly seen from both the absolute value and the relative change percentage.

By calculating the coefficient of the first-order linear equation that produces the minimum fitting error, we obtained different fitting results (Figure 9 and Table 1). For pulsed lasers of different wavelengths, the fitted straight lines exhibited different slopes and intercepts, which also confirmed that an ex vivo tissue sample has different absorption properties for different wavelengths of laser. The PA results corresponding to the 860-nm laser had a more obvious trend with temperature, which was manifested in that the PATF increased by 17% with temperature change, while the corresponding results of the 960-nm laser increased by 9%. Figure 10 shows the temperature measurement error corresponding to the two lasers, and the temperature measurement error of the 860 nm laser is significantly smaller than that of the 960 nm laser, and their *RMSE*s were 0.78 °C and 0.92 °C, respectively. It can be seen that the higher the fitting coefficient $R^{2}$, the greater the accuracy of the temperature measurement.

Although the 860 nm laser showed better temperature measurement accuracy than the 960 nm laser, the fitting determination coefficients did not exceed 0.9. To obtain higher measurement and control accuracy, a temperature calibration and measurement experiment based on multi-wavelength information was performed. As shown in Figure 11, a temperature calibration plane was constructed to fit the data points of the PATF information provided by the two lasers. The fusion of temperature measurement information leads to higher measurement accuracy (Table 2 and Figure 12). It can be seen that the fitting determination coefficient $R^{2}$ and *RMSE* were 0.96 and 0.56 °C, and a reduction of 27.5% and 38.9% in *RMSE* compared to a single wavelength of 860 and 960 nm. 

Through the multi-wavelength PA temperature calibration and measurement experiment, we tested the feasibility of the MPA-PTT system for PA temperature measurement in vitro, and we verified that the temperature measurement method based on multi-wavelength PA information has greater accuracy than that based on single-wavelength PA information through a comparative experiment. Because thermometry precision is one of the significant factors affecting temperature control accuracy, it also provides a potential opportunity to further improve the thermal dose control accuracy of PTT.

### 4.2. Closed-Loop Photothermal Therapy

The closed-loop photothermal therapy experiment was performed to verify the ability of the MPA-PTT system to regulate therapy temperature. The swine blood sample was first kept at 33 °C for 60 s to simulate the stable temperature and this temperature was also established as the base temperature for PPT. Before PTT, we collected the PA and ultrasonic images of the sample to obtain its structure and light absorption information (Figure 13). In particular, differences in the light absorption properties of the samples at different wavelengths of the laser cause the two PA images to exhibit different characteristics. The ROI area is a rectangle with side lengths of 12 and 10 mm, and we set a sampling line in the PA imaging area to more intuitively observe the impact of the temperature change on the PA images. Multimodal imaging results can provide more auxiliary information for PTT, and a more accurate selection of the treatment area can reduce thermal damage to normal tissues.

During PTT, the desired treatment temperature was set at 42 ℃ and stabilized for 80 s to meet the MPTT requirements [11]. The values of the initial control parameters were obtained using the Ziegler–Nichols method and were optimized via manual adjustment on site [27,28]; the control parameters used in the final experiments are shown in Table 3.

To verify the improvement effect of multi-wavelength PA information on temperature control, the control group experiment based on a single wavelength laser was also performed, and the measurement results of thermocouple were recorded during the control process. Temperature tracking curves are shown in Figure 14. It can be seen that, although the single-wavelength temperature control algorithm based on 860 nm and 960 nm lasers can also follow the desired temperature, the temperature-following situation shows greater volatility, that is, the steady-state performance of the temperature control algorithm is poor. It is worth mentioning that the measurement results for the 860 nm wavelength are better than those for the 960 nm wavelength, which is also in line with the temperature measurement accuracy results. All of the control results based on PA information slightly lagged behind the results of the thermocouple in transient performance, which was caused by the time-consuming operation of calculation. This slight lag did not affect the final steady-state control results; therefore, it is considered acceptable for PTT. Figure 15 shows the change in the PA intensity values on the sampling line in Figure 13 during heating; it also shows that the linear PA image reconstruction method transfers the temperature correlation of the initial PA sound pressure to the PA image.

To further analyze the temperature control capability of the MPA-PTT system, we calculated the steady-state error curve of the single-wavelength method and multi-wavelength method with reference to the thermocouple measurement results (Figure 16). Since the temperature control problem in PTT is a process control problem, the error curves only show the steady-state part of the system after 100 s of operation. To better show the performance differences between different methods, the *RMSE* is listed in Table 4, which shows that the multi-wavelength method integrates the temperature information of different lasers well and improves the temperature control accuracy. Compared to the single wavelength method, its temperature control *RMSE* is reduced by 29.2% and 40.3% (860-nm and 960-nm), respectively, and the temperature control *RMSE* is within 0.7 °C.

The ROI temperature imaging results were obtained as shown in Figure 17a when the sample temperature control reached the stable stage and the signal oscillated upward and downward around the desired temperature. The temperature imaging results clearly show the temperature distribution of the ROI. Figure 17a shows that the highest temperature in the temperature image was 41.7 °C, which was very close to the expected treatment temperature of 42 °C, and there is no local hyperthermia, which reduces the possibility of overtreatment. The results of the multimodal temperature and ultrasound imaging show the structure and functional information of the sample (Figure 17b), which has the potential to provide additional auxiliary information about the key treatment for PTT.

Through the closed-loop PTT experiment, we verified the closed-loop therapy temperature control function of the MPA-PTT system. The accuracy improvement of the temperature control method based on multi-wavelength PA information compared with the method based on single-wavelength PA information was verified through a comparative experiment. The closed-loop PTT mode guided by multimodal images can reduce the possibility of overtreatment, and steady-state temperature control results also demonstrate the potential of the proposed system for improving the safety of PTT.

### 4.3. Anti-Interference Ability Test

The stability and safety of the treatment system are crucial during PTT, which directly affects the effectiveness of treatment. The robustness of the closed-loop control system refers to the ability to maintain a stable state without changing. To test the robustness of the proposed system, after the system reached the stable stage, we used another CW laser to heat the sample with an energy density of 8 mJ/cm^2^ for an additional 20 s as an external disturbance. The results of the temperature control are shown in Figure 18. After the disturbance was introduced, the temperature-following curves both showed an oscillation phenomenon, which means that the system was no longer in a stable state and could not guarantee working stability. The MPA-TCM could quickly return to a stable state within 20 s after the introduction of disturbance, and the temperature-following curve did not oscillate. In particular, the maximum following error of this method during the adjustment process after adding disturbance is 1.4 °C, while the maximum errors of the single-wavelength methods (860 and 960 nm) were 2.4 °C and 4 °C, respectively, which were 71.4% and 185.7% larger than the results of MPA-TCM. Higher robustness can effectively reduce the possibility of unpredictable localized high temperature in PTT, and can also avoid the serious consequences of tissue carbonization.

Through the anti-interference ability test, we tested the robustness of the MPA-PTT system. The experimental results show that the fusion of multi-wavelength PA information improves not only the control accuracy but also the anti-interference ability of the system. Even when unknown interference was generated during treatment, the system demonstrated the ability to adjust itself to a stable state. This ability can effectively reduce the possibility of unpredictable local high temperature during treatment, thus avoiding serious consequences such as tissue carbonization.

## 5. Discussion

The tissue composition of the area to be treated is generally complex in PTT. However, the Grüneisen coefficients of water-based tissues and lipid-based tissues show opposite trends in temperature dependence. Therefore, it is necessary to carry out specific research based on more types of tissues and establish rich databases and theoretical models. The introduction of specific nanoprobes for the tissue to be studied is also expected to solve this problem, while only the tissue to be monitored absorbs the laser energy and generates thermalelastic transformation to induce PA signals; this will further reduce the problem caused by the complexity of the tissue. In addition, a linear array probe with a center frequency of 7.5 MHz is used in this paper, and the superior near-field view makes it more suitable for application scenarios of superficial tissues. In the future, more forms of probe will be based on the application. Scenes are added to the system. The handheld form makes the application of the integrated probe more flexible, and the introduction of the motion control system will make the image and temperature measurement process more automatic and precise. In the future, more forms of probe will be added to the system, including an annular array or a single-element probe according to the application scenarios. The current handheld form makes the application of the integrated probe more flexible, and the introduction of the operation control system will make the image and temperature measurement process more automatic and accurate.

Although increasing the number of laser wavelengths used for measurement can further improve the control accuracy, the increase in the amount of calculation brings control hysteresis, which will increase the difficulty of temperature control. In particular, there are two sources of temperature control errors. One part comes from the PA temperature measurement error, and the other part comes from the control error of the control algorithm itself. If the two parts of the error can be coordinated and dealt with well, this may improve the control performance and make its application in PTT more reliable. For the temperature measurement error, neural network theory can be used to calibrate different biological tissues to further improve the accuracy of temperature measurement. For the control error, fuzzy control or other adaptive control algorithms are used to further reduce the control error and improve the dynamic and static performance of the system. In essence, the design proposed in this paper belongs to the dark-field illumination design. Although this kind of design is more concise and convenient, the design concept based on bright-field illumination can further improve the imaging quality of the system.

During PTT, changes in tissue properties and blood perfusion caused by heating will become the key factors affecting temperature measurement. The target temperature of MPTT will be controlled below 43 °C, reducing the possibility of drastic changes in the light absorption properties of the tissue, and more accurate coupling relationships can be obtained after repeated calibrations. In addition, living tissue-based calibration models should be investigated to consider the effects of changes in blood perfusion or other tissue properties, and to build a rich calibration database. The depth of treatment is also one of the key factors of PTT. The multi-wavelength photoacoustic temperature feedback based PTT method recommended in this manuscript is based on PA and PTT technology. PA technology combines the high penetration depth of ultrasonic technology, and the penetration depth of millimeters to centimeters can be achieved according to different types of ultrasonic transducers. As for PTT technology, a higher laser wavelength leads to a higher penetration depth. In addition, the introduction of nanophotosensitizers can further improve the light absorption efficiency of the lesion. In the proposed MPA-PTT system, we designed the flexible hand-held PTT probe, and the multiwavelength treatment laser uniformly irradiates the area to be measured through fiber bundle; combined with the multimode imaging function of the system, the penetration depth of cm level can be achieved by manually adjusting the longitudinal height. In future work, a mechanical transmission mechanism to drive the lens to realize automatic adjustment of laser irradiation angle will be considered, and the positioning function of ultrasonic and photoacoustic imaging combined with the automatic adjustment of the PTT laser will have the potential to make PTT more flexible and convenient.

## 6. Conclusions

Precise temperature control of biological tissue based on PA thermometry is applicable to tumor PTT. Based on the experimental results, the proposed multi-wavelength photoacoustic temperature feedback based PTT method can be used to: (1) control the therapy temperature of the ROI with an accuracy of <0.7 °C in PTT; (2) noninvasively estimate the temperature changes with an accuracy of < 0.6 °C in PTT; (3) complete real-time temperature imaging for tissue; and (4) maintain strong robustness to handle external interference and measurement noise, which can ensure the safety and effectiveness of PTT. Compared to the single-wavelength measurement results at 860 and 960 nm, the temperature measurement errors are reduced by 27.5% and 38.9%, respectively, and the temperature control errors are reduced by 28.8% and 40.3%, respectively. In conclusion, this method has the potential to increase the safety and accuracy of MPTT.

## Figures and Tables

**Figure 1 pharmaceutics-15-00555-f001:**
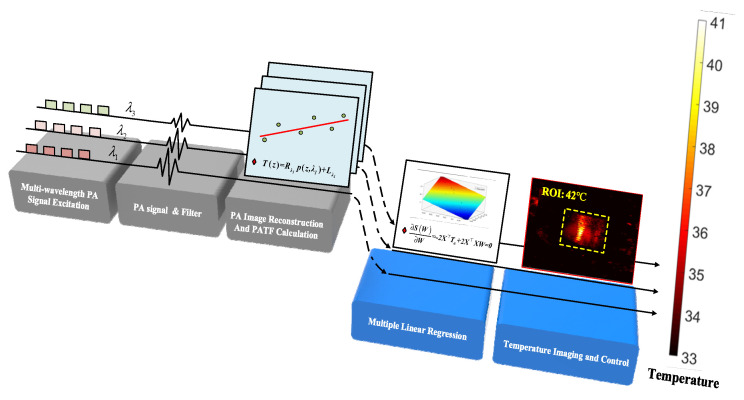
The MPA-TCM flow chart.

**Figure 2 pharmaceutics-15-00555-f002:**
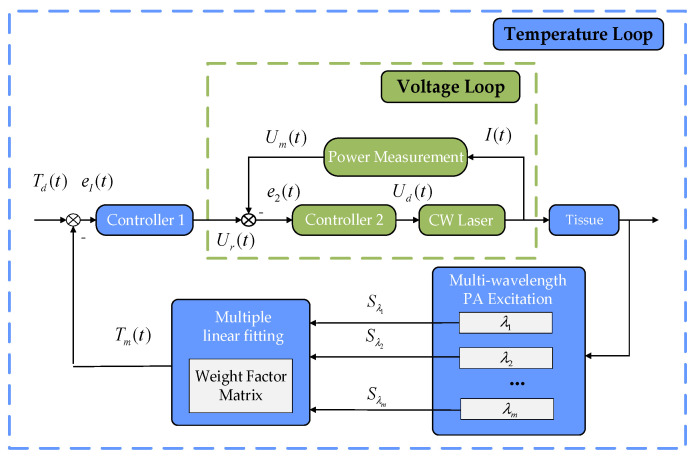
The cascade PID control strategy.

**Figure 3 pharmaceutics-15-00555-f003:**
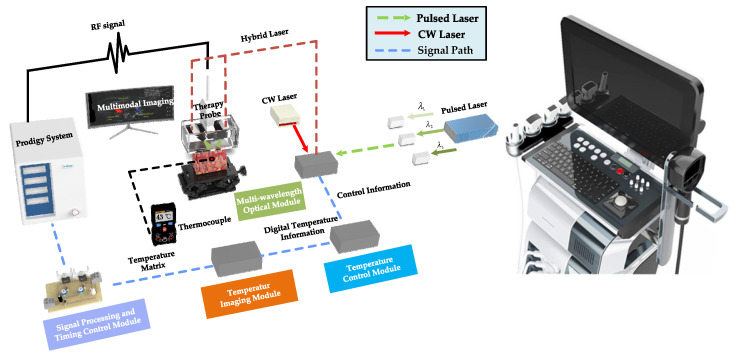
Structural diagram of the MPA-PTT system.

**Figure 4 pharmaceutics-15-00555-f004:**
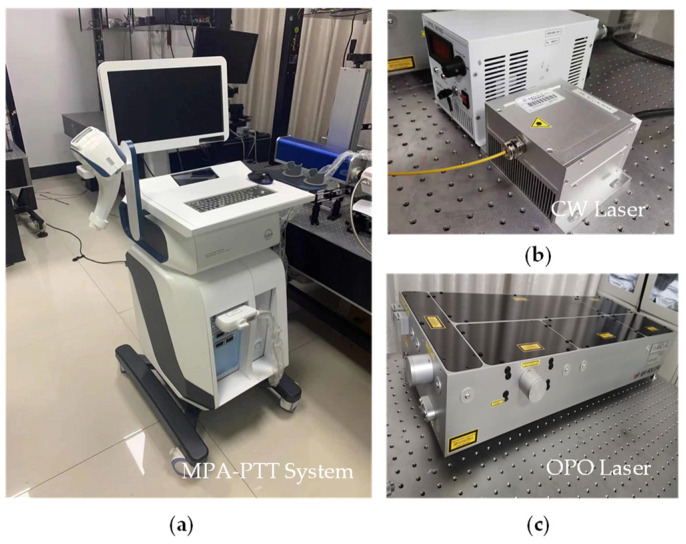
Light source and temperature control module of the MPA-PTT system: (**a**) CW pulse; (**b**) optical parametric oscillator; (**c**) temperature control module.

**Figure 5 pharmaceutics-15-00555-f005:**
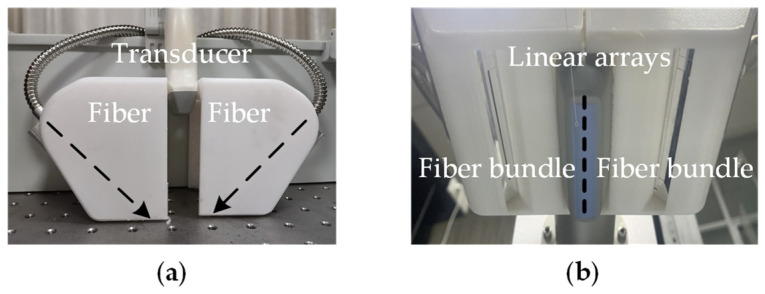
Integrated probe: (**a**) lateral view; (**b**) light exit.

**Figure 6 pharmaceutics-15-00555-f006:**
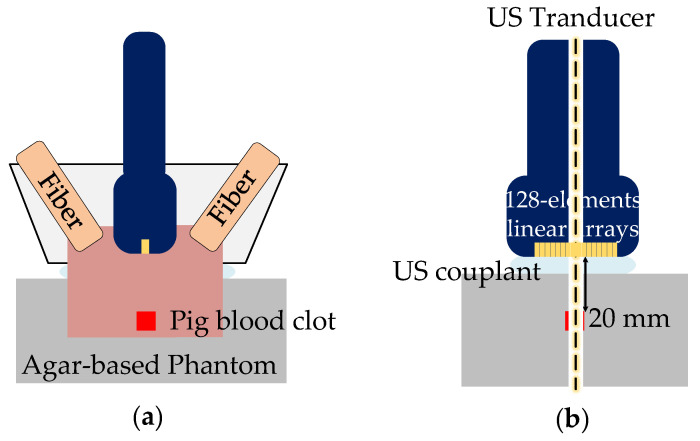
Schematic of the experimental setup: (**a**) front view; (**b**) side view.

**Figure 7 pharmaceutics-15-00555-f007:**
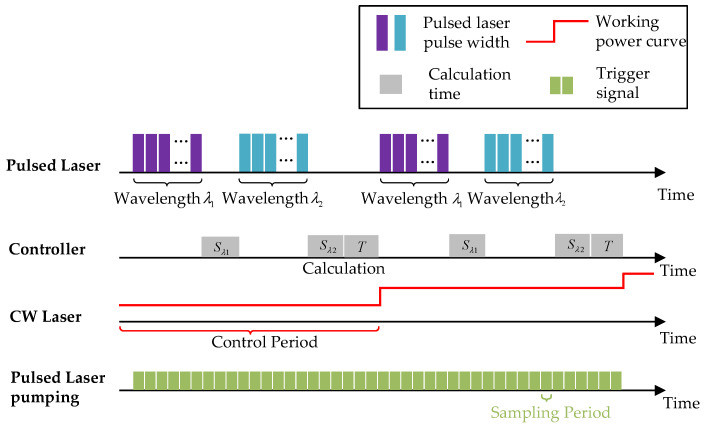
The working sequence diagram of the MPA-PTT system.

**Figure 8 pharmaceutics-15-00555-f008:**
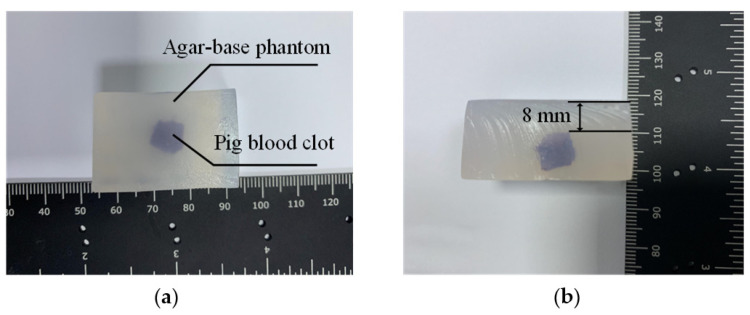
Ex vivo swine blood sample: (**a**) top view; (**b**) side view.

**Figure 9 pharmaceutics-15-00555-f009:**
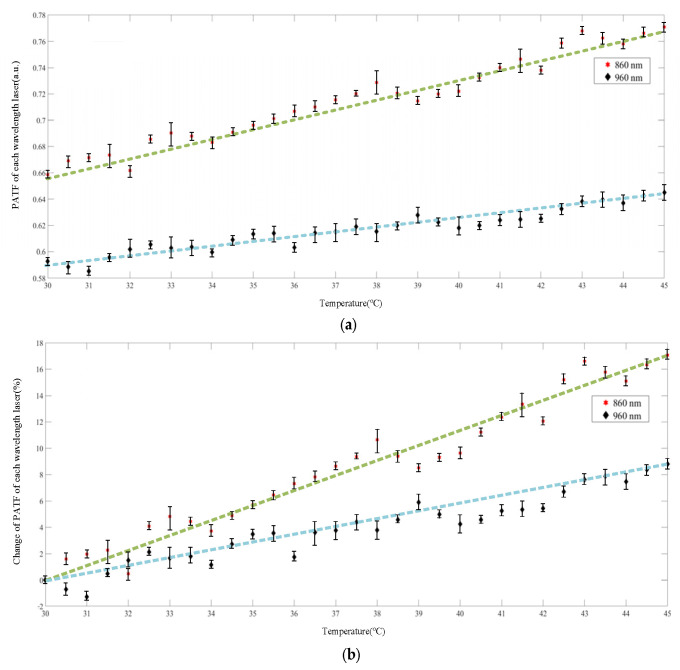
Temperature calibration results for the swine blood sample: (**a**) absolute calibration; (**b**) calibration percentage of relative change.

**Figure 10 pharmaceutics-15-00555-f010:**
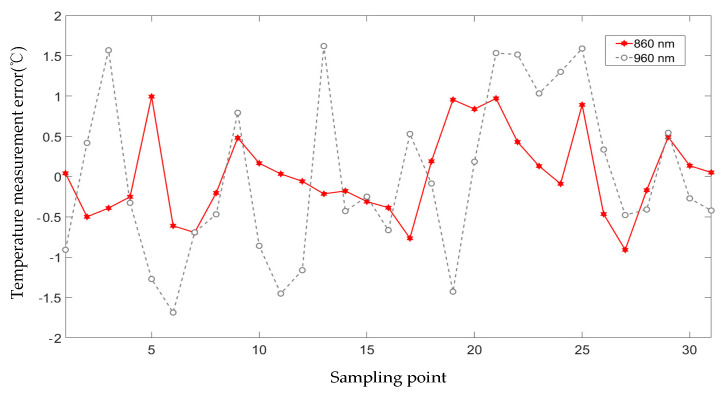
Temperature measurement error for different laser wavelengths.

**Figure 11 pharmaceutics-15-00555-f011:**
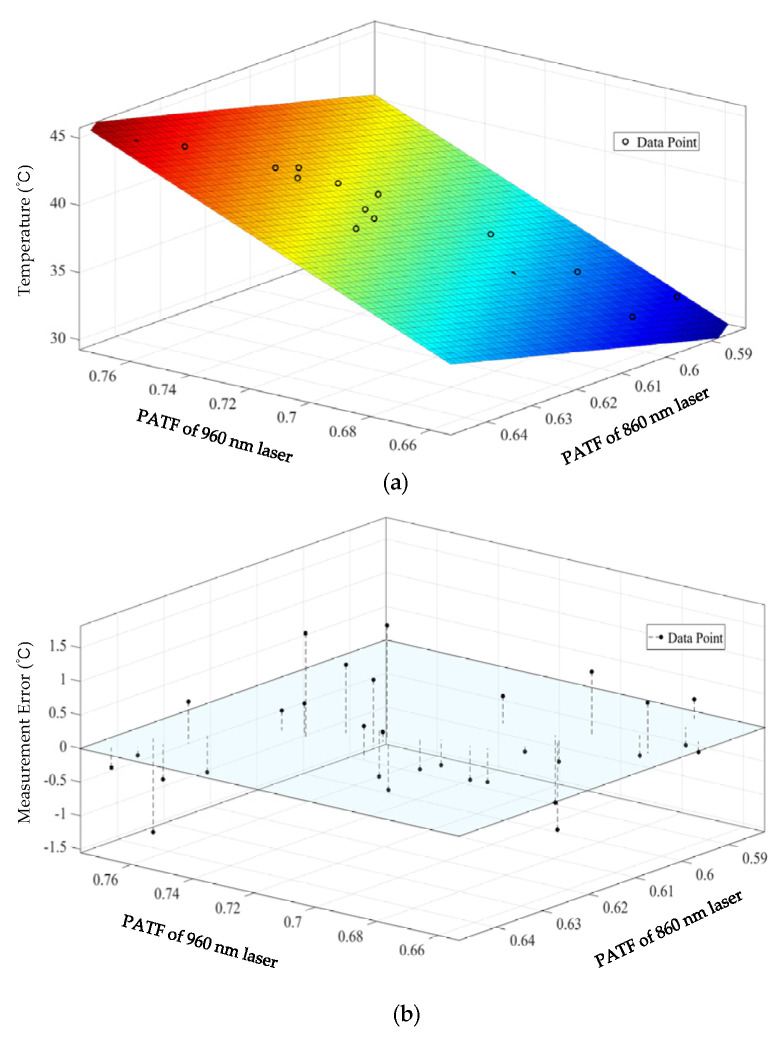
Multi-wavelength temperature calibration results: (**a**) three-dimensional plane fitting; (**b**) three-dimensional view of residuals.

**Figure 12 pharmaceutics-15-00555-f012:**
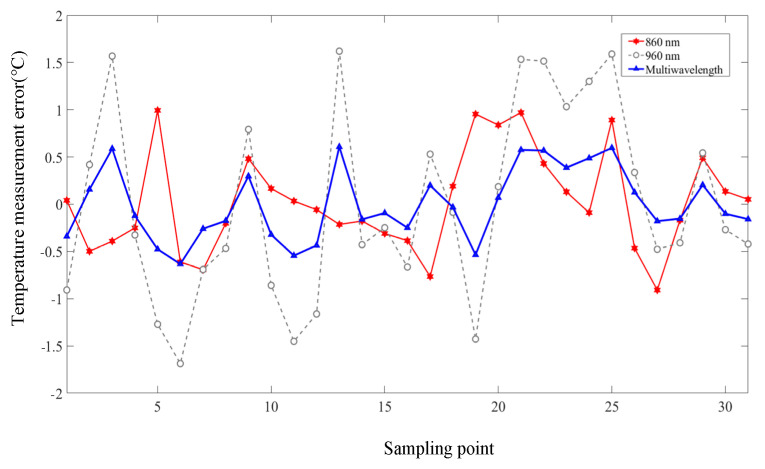
Comparison of temperature measurement errors of the multi-wavelength and single-wavelength temperature measurement algorithm.

**Figure 13 pharmaceutics-15-00555-f013:**
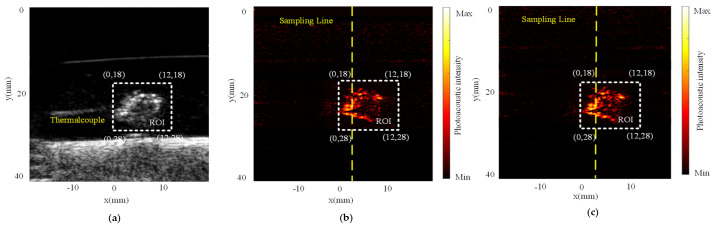
Ultrasound and PA images of the swine blood sample at 33 °C: (**a**) ultrasound image; (**b**) 960-nm laser PA image; (**c**) 860-nm laser PA image.

**Figure 14 pharmaceutics-15-00555-f014:**
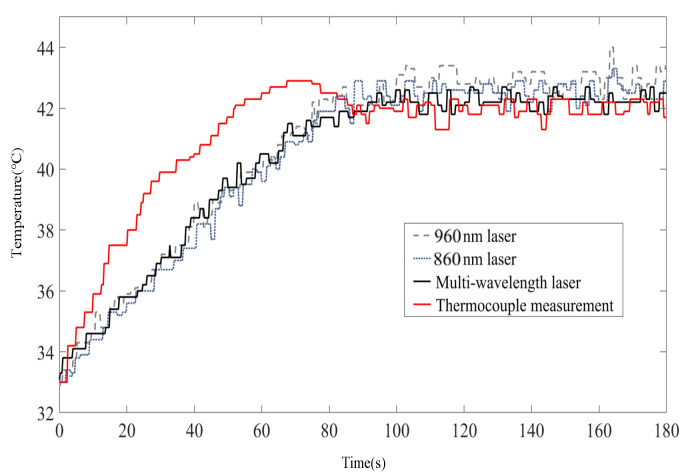
Temperature-following curves for the swine blood sample.

**Figure 15 pharmaceutics-15-00555-f015:**
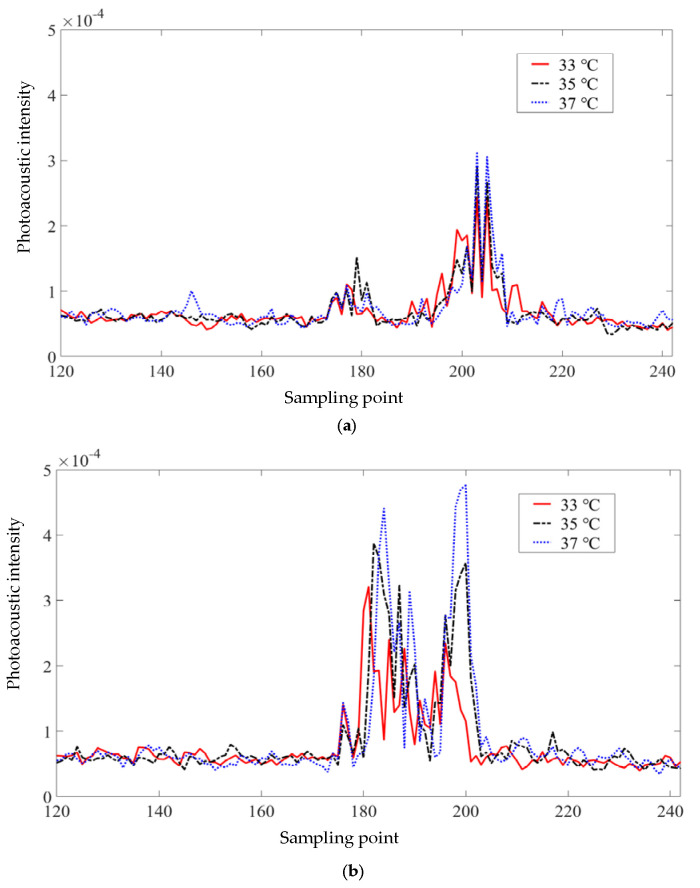
Intensity values on the sampling line of PA images at different temperatures excited by lasers of different wavelengths: (**a**) 960 nm laser; (**b**) 860 nm laser.

**Figure 16 pharmaceutics-15-00555-f016:**
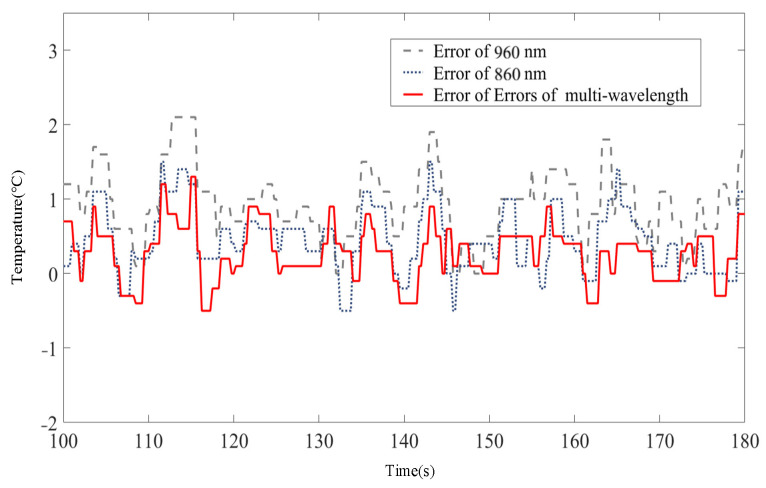
Steady-state temperature-following error curves.

**Figure 17 pharmaceutics-15-00555-f017:**
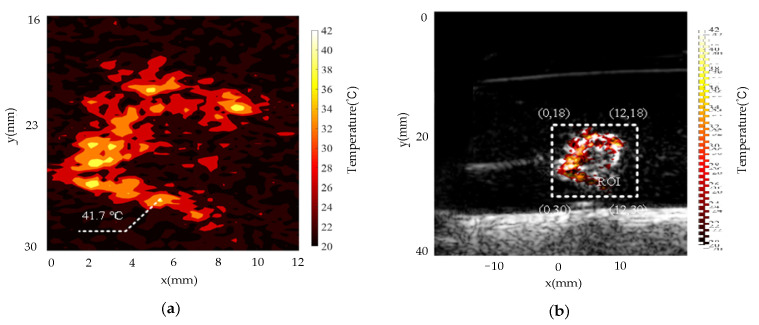
Imaging results when the MPA-PTT system reaches a temperature stable state:(**a**) temperature imaging of the ROI; (**b**) ultrasonic temperature dual-mode imaging map.

**Figure 18 pharmaceutics-15-00555-f018:**
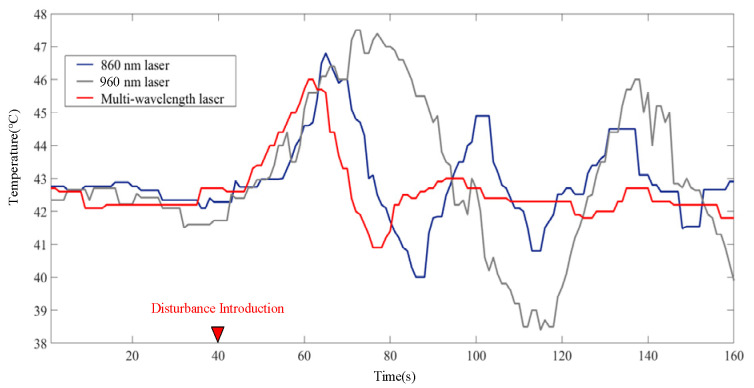
Temperature control curve of the MPA-PTT system under external disturbance.

**Table 1 pharmaceutics-15-00555-t001:** Calibration results of the PATF and temperature for the 860 nm and 960 nm lasers.

Wavelength	Equation	$R^{2}$	*RMSE*
860-nm ($\lambda_{1}$)	$T=133.31\cdot S_{\lambda_{1}}-57.86$	0.89	0.78
960-nm ($\lambda_{2}$)	$T=275.72\cdot S_{\lambda_{2}}-132.36$	0.86	0.92

**Table 2 pharmaceutics-15-00555-t002:** Temperature measurement calibration accuracy of the multi-wavelength temperature measurement algorithm.

Equation	$R^{2}$	*RMSE*
$T=93.71\cdot S_{\lambda_{1}}+88.22\cdot S_{\lambda_{2}}-83.85$	0.96	0.56

**Table 3 pharmaceutics-15-00555-t003:** Control parameters of the temperature loop and voltage loop of the system.

Temperature Loop Proportional Coefficient	Temperature Loop Differential Coefficient	Voltage Loop Proportional Coefficient	Voltage Loop Integral Coefficient
3	1.2	0.89	0.78

**Table 4 pharmaceutics-15-00555-t004:** Results of temperature control errors.

Equation Excitation Wavelength	*RMSE*
860-nm	0.96
960-nm	1.14
Multi-wavelength (860- and 960-nm)	0.68

## Data Availability

No new data were created.

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
