# Peer review of "Multi-Wavelength Photoacoustic Temperature Feedback Based Photothermal Therapy Method and System"

_pharmaceutics, 2023, doi:10.3390/pharmaceutics15020555_

Round 1

Reviewer 1 Report

From my point of view, the article should be entirely rewritten before further analysis. 

Specifically, the authors should reorganise the article to have a clear structure. Different methodologies are mixed in other places. The experimental section is not clear at all, at least for me. 

There are a lot of questions about the model description; specifically for me, it is not clear where N disappeared between Eq (3) and (4). Also, there is a question concerning Eq. 1 and 2. 

Author Response

Thank you for your hard work! Please see the attachment.

Reviewer 2 Report

This report aims to improve therapy accuracy and safety , a multi-wavelength photoacoustic (PA) temperature feedback based PTT method and system. Although the study is interesting and promising, the study suffers from major experimental design flaws.

In the case of experimental output, authors should give additional information how they obtained PA intensity values  in Figure 14 which they used to generate image in Figure 13. For example, the shape of photoacoustic signal is important for two different temperature, as well as two different wavelengths and the characteristic of signals should be compared. The intensity of PA signal should not use as output for this experiment, or at least the authors should explain how they generated the image,  for example, if the authors use the energy signal under PA signals, they should indicate this.

Additionally, the  spot size of the laser is also important based on combination of  photothermal and photoacoustic effect. What is the diameter of the fiber?

What is the depth of blood locations into agar? Based on in vivo experiment, the depth is important to reach to tumor environments. The authors criticize Ultrasonic thermometry technology according to  requirements of real-time measurement and detection depth, however, they did not give information about their depth capability or how that change their system based on different depths, how they know that their measurement precision is high enough to implement for in vivo applications??

In the discussion part, when the authors make some conclusion to improve the results, they should be careful so as not to mislead the readers. For  instance, why the higher repetition frequency reduce the influence of the measurement? Is the *below sentence is general in this field, or is it valid for the system in this work? Or is it related to the specific design? I think, depending on the electronic part of the system, this sentence can change. Without showing noise level, or compare two different PRF, the sentence is not completely right

*The below sentence:

In system design, the use of pulsed lasers with higher repetition frequency can 453 further reduce the influence of measurement noise to achieve higher temperature meas- 454 urement accuracy faster feedback response.

Finally, the other important problem in this interesting work is the usage of real-time measurement. As a summary, OPO laser in the work was used as reader based on photoacoustic. However, the repetition rate is 10 Hz, meaning that the acquisition time of image is  low. Why the authors claim that this is real time measurement.

Author Response

(The authors gave the same response as above.)

Round 2

Reviewer 1 Report

The current version of the article is better. However, some incorrectnesses are still presented there. One gets the impression from the introduction that the main mechanism of the PA signal formation is dependence of Grüneisen parameter on the temperature. But it is not like this - thermalelastic transformation is the principle mechanism. Authors, should rebuild their considaration from this fact.

Author Response

Your good advice was very much appreciated, and please see the attachment.

Reviewer 2 Report

Thanks for your all reply. 

Round 3

Reviewer 1 Report

I think that the article canbe accepted in the presented form